# Web-Based Physical Activity Interventions to Promote Resilience and Mindfulness Amid the COVID-19 Pandemic: A Pilot Study

**DOI:** 10.3390/ijerph20085463

**Published:** 2023-04-11

**Authors:** Michele W. Marenus, Ana Cahuas, Dianna Hammoud, Andy Murray, Kathryn Friedman, Haley Ottensoser, Julia Sanowski, Varun Kumavarel, Weiyun Chen

**Affiliations:** 1School of Kinesiology, University of Michigan, Ann Arbor, MI 48109, USA; 2Department of Psychology, University of Michigan, Ann Arbor, MI 48109, USA

**Keywords:** virtual intervention, aerobic and resistance training, mindful exercises, resilience

## Abstract

College students faced unique challenges during the COVID-19 pandemic. Implementing a physical activity intervention can help support the physical and mental health of college students. The purpose of this study was to examine the effectiveness of an aerobic–strength training exercise intervention (*WeActive*) and a mindful exercise intervention (*WeMindful*) in improving resilience and mindfulness among college students. Seventy-two students from a major public university in the Midwest participated in a two-arm experimental study over the course of ten weeks. One week before and after the 8-week interventions, participants completed the Five-Facet Mindfulness Questionnaire (FFMQ-15), Connor Davidson Resilience Scale (CD-RISC-10), and demographic and background questionnaire via Qualtrics. Both groups also participated in bi-weekly Peer Coaching sessions, which utilized reflective journaling and goal-setting exercises. ANCOVA showed a significant main effect of time for total mindfulness score (*F* = 5.177, *p* < 0.05, *η^2^* = 0.070), mindfulness Acting with Awareness (*F* = 7.321, *p* < 0.05, *η^2^* = 0.096), and mindfulness Non-Judging of Inner Experience (*F* = 5.467, *p* < 0.05, *η^2^* = 0.073). No significant main effect of group and interaction effects of time with group were observed for the total mindfulness and the five facets of mindfulness as well as resilience. In addition, no significant main effect of time for resilience was found. We conclude that aerobic–strength exercises and mindful yoga exercises, together with reflective journaling, may be effective in increasing mindfulness in the college population.

## 1. Introduction

The COVID-19 pandemic has changed day-to-day life around the globe. The unprecedented nature and spread of the disease resulting in shutdowns and restricted socializing all contribute to feelings of uncertainty and stress. On top of the risk of viral infection, decreased mental health poses a secondary public health crisis. In particular, college students have faced unique challenges such as managing a loss of in-person classes, reduced socialization, a transition to remote learning, and, for many, having to leave campus. In response to the COVID-19 pandemic, students have reported psychological stress, difficulty coping, and a desire to return to campus [1]. The pandemic has also harmed college students’ health behaviors, including worsened sleep patterns, increased sedentary and screen time, and decreased physical activity [2]. These impacts are cause for concern because college can be a stressful time and an arduous transition under normal circumstances, and mental health issues are on the rise in the student population [3]. Universities must consider both the physical and mental impacts of the pandemic on students’ well-being in order to support them throughout the ongoing transition back to campus and in-person learning. Specifically, cultivating resilience and mindfulness across campuses may be a promising way to fight against the negative impacts of the pandemic and promote both physical and mental health.

Resilience is a highly adaptive trait that describes one’s ability to overcome hardships and maintain a sense of purpose, health, and growth, despite setbacks and difficult life events [4]. Backed by decades of research, Seligman [5] cites optimism as a key contributor to resilience. Thinking optimistically can prevent depression, anxiety, and even giving up after failure, a cornerstone of resilience [5]. Seligman’s research has been supported by a large body of literature demonstrating the benefits and roots of resilience. Studies have found that resilience is associated with both improved academic achievement and mental health among undergraduate students [6]. Specifically, higher resilience is associated with lower depression in adolescents [7]. A cross-sectional study of 303 college students in the Philippines during COVID-19 found that resilience protects against loneliness [8]. A similar study finds higher resilience to be associated with lower anxiety in nursing students in Israel during the pandemic [9]. Research has shown that resilience is good for overall well-being, can promote the healing process, and anyone can develop resilience in his/her own life [10]. 

In addition to the documented benefits, physical activity interventions have shown success in increasing resilience, including web-based programs [11,12]. Interventions using moderate to vigorous physical activity as well as mindfulness training have increased resilience in the college student population [13,14]. Examining physical activity and resilience in adolescents, a review proposes that exercise and physical fitness builds resilience by strengthening neural pathways associated with self-regulation [15]. Similarly, a review on exercise and resilience concludes that regular physical activity throughout several stages of life can lead to a more resilient brain by protecting it from cognitive impairment and brain pathology [16]. Yoga may have similar effects. Brown and Gerbarg [17] suggest that practicing yoga with mindful breathing can strengthen resilience to stress and alleviate suffering. A meta-analysis finds that a majority of mindful exercise interventions, including meditation, yoga, tai chi, and qigong, improve resilience among maltreated youth [18]. A randomized controlled trial using an 8-week mindfulness course reports that mindfulness increases resilience by decreasing distress and increasing well-being in 309 university students, compared to 307 students in the control group [19]. Another resilience program intervention, including goal-building, mindfulness, and resilience training implemented through orientation courses, demonstrates improvements in depression, perceived stress, emotion regulation, and mindfulness for 252 first-year college students over the course of a semester [20]. Overall, resilience is a key contributor to thriving that can be cultivated. 

Similar to resilience, mindfulness is an adaptive practice that decreases stress and aids with emotion regulation [21]. The practice of mindfulness entails observation of the present moment, including awareness of one’s environment, thoughts, and feelings without judgment [22]. Although there are several techniques for reaching a mindful state, mindfulness typically has two components: observation and acceptance of the present moment. A randomized control trial of 137 adults comparing monitoring and acceptance, monitoring only, and control group finds that acceptance of the present moment is a key component in decreasing stress and increasing nonjudgment [23]. Both seated and motion-based mindfulness have been found to increase openness, which is associated with improved ability to cope with stress [24]. A study on movement-based classes including Pilates, Taiji Quan, or GYROKINESIS^®^ finds an increase in mindfulness and improvements in mood, perceived stress, and sleep among 166 college students in the three groups [25]. A similar randomized controlled trial assessing a mindfulness training program called Koru with 90 college students demonstrates increased mindfulness, lowered perceived stress, improved sleep, and higher self-compassion [26]. A meta-analysis of 20 studies reports that mindfulness-based stress reduction intervention show promise in helping both clinical and nonclinical populations with coping [27]. Yoga is another evidence-based practice that facilitates mindfulness. Higher yoga involvement is associated with increased mindfulness as well as improved mood [28]. An 8-week biweekly Vinyasa yoga intervention with a sample of 20 college students shows an increase in positive affect and a decrease in negative affect [29]. Another 8-week weekly yoga intervention with a sample of 66 university students finds increased mindfulness as well as decreased stress, worry, and depression post-intervention [30]. Although the benefits are clear, one critical evaluation of mindfulness warns that “misinformation and poor methodology” exist in recent studies, and recent mindfulness trends may take away from the intended practice [31]. All considered, the benefits of mindfulness for overall well-being are widely documented. 

The effects of web-based interventions for physical and mental health are yet emerging. It is worth investigating the realm of virtual programs seeing as they have become commonplace for many types of meetings and classes throughout the pandemic. Therefore, we designed and implemented the web-based aerobic and strength exercise intervention named *WeActive* and a mindful exercise intervention named *WeMindful* [32,33]. This study specifically focuses on examining the effects of the *WeActive* and *WeMindful* interventions on resilience and mindfulness among college students over the course of 8 weeks. We hypothesize that both *WeActive* and *WeMindful* interventions will increase resilience and mindfulness over time. Additionally, we hypothesize that *WeMindful* will show a greater increase in the two study outcomes compared to *WeActive*. This study is significant because it explores a potential and accessible platform to support students’ physical and mental well-being. 

## 2. Materials and Methods

### 2.1. Participants and Study Design

We recruited the participants using the university’s targeted email request system, fliers, an Instagram page, and the university’s Learning Management System—Canvas Dashboard. To be eligible for participation in the study, each participant needed to meet the following criteria: current enrollment as either an undergraduate or a graduate student, ability to participate in all assessments and intervention activities, and completion of the consent form. Participants were excluded if they were under 18 years of age, or if they had injuries or illness prevented them from being able to perform exercises regularly. Eligible participants completed the consent form prior to participation. The University Institutional Review Board of Health and Behavioral Sciences approved this study (HUM00189120/Ame00107415). 

We used G* Power 3.1.9.7 software to calculate the study sample size with an effect size, Cohen *d* = 0.20, on college students’ state mindfulness, 2-tailed with α level of 0.05, and a power of 0.85 using repeated measures ANCOVA. The results of the power analysis showed that the total required sample size of the study was 48 with an actual power of 0.86, with 24 students in each intervention group. Based on our previous study’s adherence rate of >80% and an estimated program dropout rate of 20%, we needed to recruit a targeted sample size of 58 college students [34,35]. Our sample size of 77 participants exceeded the required sample size of 58.

This two-arm, experimental study was conducted over the course of ten weeks. One week before the 8-week interventions, each participant completed the Resilience and the Mindfulness questionnaires, self-reporting whether they had engaged in physical activity or yoga within the past three months, whether or not they had visited a psychological therapist in the past three months, as well as their overall health rating on a 5-point scale, and demographic information via Qualtrics. The participants were then randomly assigned to either the *WeActive* or *WeMindful* intervention group. Participants completed 8-weeks of intervention in their corresponding group. One week after the intervention, each participant completed the two questionnaires via Qualtrics again. 

### 2.2. Outcome Measures 

#### 2.2.1. Resilience

The Connor Davidson Resilience Scale (CD-RISC-10) is a validated and widely used 10-item scale to measure one’s ability to bounce back from stressful events, tragedy, or trauma [36]. Sample items on the questionnaire include “I can deal with whatever comes my way,” “I believe I can achieve my goals, even if there are obstacles,” and “I am not easily discouraged by failure.” The participants were asked to respond to each item on a 5-point rating scale that ranged from 1 = not true at all to 5 = true nearly all the time. The total scale score was calculated by adding the ten items. Gonzalez, Moore, Newton, and Galli [37] indicated that this ten-item unidimensional scale demonstrates internal consistency, with a Cronbach’s alpha of 0.87. In this study, the total scale had 0.759 and 0.773 Cronbach Alpha coefficients at the baseline and the post-intervention test, indicating acceptable internal consistency. 

#### 2.2.2. Mindfulness

The Five-Facet Mindfulness Questionnaire (FFMQ-15) is a 15-item measure to assess mindfulness in five facets, including Observing, Describing, Acting with Awareness, Non-Judging of Inner Experience, and Non-Reactivity to Inner Experience on a 5-point rating scale [38]. The FFMQ-15 is a short form of the 39-item FFMQ that was developed by Baer et al. [39], which originally included three items for the five facets of mindfulness. After examining the factor structure and psychometric properties of the FFMQ-15 in comparison to the original 39-item scale, Gu et al. [38] concluded that when briefer forms are used, the condensed scale is an adequate alternative measure to the original 39-item questionnaire (FFMQ-39), as no significant differences were found in convergent validity and internal consistency. Based on suggestions from Gu et al. [38], three items on the Observing facet were omitted from the data analysis of the study because previous research has regarded scores from these items before and after Mindfulness-Based Cognitive Therapy as invalid. 

Three items on each facet of mindfulness were included in the survey. An example of an item on Describing is “I believe some of my thoughts are abnormal or bad and I shouldn’t think that way”. A sample from an item on Acting with Awareness is “I do jobs automatically without being aware of what I’m doing”. An example from an item on Non-Judging of Inner Experience is “I think some of my emotions are bad or inappropriate and I shouldn’t feel them”. An exemplary item on Non-Reactivity to Inner Experience is “I tell myself. I shouldn’t be feeling the way I’m feeling”. The participants were asked to respond to each question on a 5-point rating scale, ranging from 1 = never or very rarely true to 5 = very often or always true. Items 3, 4, 7, 8, 9, 13, and 14 were reversely coded. The total scale score was the average score of the four subscale scores. In this study, the Cronbach’s alpha coefficients of the total scale at baseline and the post-intervention test were 0.685 and 0.858, respectively.

### 2.3. Intervention Conditions

#### 2.3.1. *WeActive* Intervention Group

Students in the *WeActive* intervention group engaged in two, 30-min virtual aerobic and strength exercise lessons each week. Participants were able to either attend the virtual lessons live through Zoom or watch the sessions at a later time, as they were recorded and made available on the study’s unique Canvas webpage. The exercise lessons were taught by a student instructor, who is a certified strength and conditioning specialist. The exercise instructor designed each lesson to begin with a 5-min warm-up, in which participants engaged in low-impact movements including walking and dynamic stretches. The participants then performed a mixture of aerobic and strength exercises for 20-min. Each lesson ended with a 5-min cooldown session using static stretching exercises. In teaching each lesson, the instructor used the “tell, show, do” instructional strategies. First, the instructor briefly introduced the lesson objectives to the participants. Then, the instructor demonstrated the correct techniques for performing each exercise, included modifications, and provided learning cues to help participants understand how to perform each exercise using proper form. Next, the instructor explained how to organize the practice of exercises using interval training, high-intensity interval training, or circuit training methods. Finally, the instructor led the participants to practice each exercise with specific repetitions or times with specific numbers of sets. The participants were able to ask the instructor via the chat function in Zoom if they had questions or needed modifications.

During the first four weeks of the study, the lessons focused more on strength exercises, including a range of lower-body, upper-body, and core exercises. Based on feedback provided by the participants, the second half of the intervention focused more on high-impact aerobic exercises. As the session was recorded and posted on the Canvas site, participants were encouraged to repeat and review the lessons on their own. Moreover, we sent an email out weekly to the participants as an act of positive reinforcement to keep them on track with their goals. 

#### 2.3.2. *WeMindful* Intervention Group

Students in the *WeMindful* intervention group attended two 30-min virtual mindful yoga exercise lessons per week. Participants were able to either attend the virtual lessons live through Zoom or watch the sessions at a later time, as they were recorded and made available on the study’s unique Canvas webpage. The mindful yoga exercise lessons were taught by a student yoga instructor who had more than two years of experience in the practice. The instructor designed each lesson to begin with a 5-min mindful warm-up that focused on mindful breathing exercises. Participants then engaged in a 20-min yoga flow, and each session ended with a 5-min mindfulness practice. The yoga flow consisted of 4–6 various poses, and the intensity slightly increased with each session over the course of the eight-week intervention. In teaching each session, the instructor used a tell-show-do instructional strategy to increase the participants’ confidence in their practice. The instructor demonstrated each pose with modifications, provided learning cues to remind key points of performing each pose, and then led the participants to practice each pose for a specific duration. After learning and practicing each pose, the instructor used the tell, show, do instructional strategy to teach a specific yoga flow (sequence) with specific sets. The instructor guided the mindful breathing warm-up and cool-down periods using verbal cues. As each session was recorded and uploaded onto the Canvas webpage, participants were encouraged to repeat and review the lessons on their own time. In addition, we sent an email to the participants weekly to positively reinforce their behavior and to keep them on track with their personal goals. 

### 2.4. Implementation Strategy

A 30-min live, Zoom-based Peer Coaching session was offered once every two weeks. A psychology student and a doctoral student, who had proper training in motivational interviewing, led the Peer Coaching sessions live over Zoom with the aim of guiding the participants to set goals in addition to reflecting on their successes, challenges, and adherence to the weekly meetings through bi-weekly journal prompts. The Peer Coaching leaders organized the journal prompts into four main categories: goal setting, self-reflection, self-monitoring, and social support. The journal prompts were tailored to each intervention group, *WeActive* and *WeMindful*. An example of a goal-setting question is, “What are your specific exercise goals for this week and next week?” or “What are your specific mindfulness goals for this week and next week?” An example of a self-reflection question is, “Please write down your top 3 personal strengths that will help you reach your weekly goals.” An example of a self-monitoring question is, “Are you facing difficulties reaching your goals?” An example of a social support question is, “Write a short note to a loved one telling them why they are important to you.” Moreover, participants were asked to share their thoughts and feelings towards the intervention activities, including their struggles and suggestions to improve the following week’s sessions. 

Each live Peer Coaching session was recorded and posted onto the Canvas webpage along with the journal prompts, allowing participants to engage in this component regardless of their ability to attend the Zoom meeting at the scheduled time. In addition to completing the journal prompts, all participants were encouraged to engage in the Peer Coaching component regardless of their availability to attend the live session, as the recorded session and journal prompts were posted to the Canvas webpage. During each Peer Coaching session, the leaders outlined the upcoming schedule of activities, introduced the journal prompts, and the participants then had five minutes to begin reflecting and completing the prompts. During a coaching session halfway through the intervention, we invited students to provide feedback on our sessions. We verbally asked participants open-ended questions about their experience in the intervention, including what they participants like and dislike about the program and how the intervention could be improved in the last week. Based on their input, we increased the amount of aerobic exercise in the *WeActive* group. The final Peer Coaching session of the study involved a goal-setting activity that implemented psychologist Gabriele Oettingen’s WOOP (Wish, Outcome, Obstacle, Plan) method [40]. The WOOP method provided a framework for attaining goals: the participants identified their wish, visualized their desired outcome, identified an inner obstacle, and made an action plan to overcome such an obstacle in the case that it arises. 

### 2.5. Data Analysis

Descriptive statistics (i.e., mean, std. deviation) of each outcome variable and baseline characteristics were computed. Independent sample t-tests were conducted to investigate whether significant differences in the demographic and outcome variables existed between the two intervention groups. A two-way repeated measures analysis of covariance (ANCOVA) was conducted to examine the effects of the interventions on the outcome variables by group over time, while statistically controlling for a covariate (pre-test seeing a therapist three times a month) due to its baseline significant difference between the two groups. The within-subjects factor was time (baseline v. post-test) and the between-subjects factor was intervention group (*WeActive* vs. *WeMindful*). All data analyses were computed using IBM SPSS 27 (IBM, Armonk, NY, USA) at the significance level of *p* < 0.05.

## 3. Results

### 3.1. Baseline Characteristics

Participants in this study were 72 students from a major public university in the Midwest. There were 63 cis-gendered females (85%), 7 cis-gendered males (10%), and 4 gender non-conforming (TGNC) people (6%). Of the 72 students, 65% identified as Caucasian, while the remaining 45% identified as either African American, Asian, Native, or Multiracial. The complete demographic information is shown in Table 1. There were 77 total participants in this intervention that completed both the baseline and post-test assessments. One hundred and two participants were measured at baseline and 25 of those participants did not complete the post-test assessment. Of the 77 total participants who completed both pre- and post-test data, five participants were excluded from the data set due to missingness. The missing data were screened using listwise deletion.

Table 2 presents baseline and post-test scores by group and for the total sample. As seen in Table 2, the mean baseline score for resilience of the total sample was 25.76. This score indicates slightly below-average resilience, as the mean score for the general US population was found to be 31.81 [41]. The total baseline mindfulness mean score for the total sample was 33.43, indicating moderate mindfulness, as scores on the FFMQ-15 range from 15 to 75. This is considered below average for this population, as the mean score for college students has been found to be 47.67 in past research [42]. Figure 1 and Figure 2 display the distribution of scores on the FFMQ-15 over time by group. 

As seen in Table 3, independent sample t-tests revealed a significant difference in visiting a therapist three times a month between the two groups (t = −2.66, df = 76, *p* = 0.000). The results indicated the *WeMindful* group had a significantly higher frequency of visiting a therapist three times a month in the past month compared to the *WeActive* group. 

### 3.2. Intervention Effects on Resilience and Mindfulness

Table 4 presents the results of the ANCOVA repeated measures. As seen in Table 4, there was no significant main effect of time or group for resilience Furthermore, there was no significant interaction between time and group for resilience when controlling for individuals who saw a therapist three times a month. 

As seen in Table 4, a significant main effect of time was observed for total mindfulness score (*F* = 5.18, *p* < 0.05, *η^2^* = 0.070), mindfulness Acting with Awareness (*F* = 7.32, *p* < 0.05, *η^2^* = 0.096), and mindfulness Non-Judging of Inner Experience (*F* = 5.47, *p* < 0.05, *η^2^* = 0.073). These results indicated that both groups experienced an increase in total mindfulness, mindfulness Acting with Awareness, and mindfulness Non-Judging of Inner Experience after the 8-week intervention (See Figure 2). 

No significant main effect of group was observed for any sub-scales and the total scale of mindfulness, indicating no significant differences in these outcome variables between *WeActive* and *WeMindful* regardless of time. Furthermore, there was no significant interaction of time and group for any sub-scales and the total scale of mindfulness, indicating there were no significant differences on these outcome variables between *WeActive* and *WeMindful* over the course of the 8-week intervention. 

## 4. Discussion

This study aimed to examine and compare the effectiveness of *WeActive* and *WeMindful* interventions on resilience and mindfulness in college students. One promising result of our study was that both 8-week *WeActive* and *WeMindful* interventions resulted in an increase in total mindfulness, mindfulness Acting with Awareness, and mindfulness Non-Judging of Inner Experience. These results indicate that both aerobic–strength exercises and mindful yoga exercises, accompanied by the goal-setting and reflective journaling used in Peer Coaching sessions, are effective approaches to increasing mindfulness. 

It is evident that there is not one way to increase mindfulness, as seen in the present study and previous literature. This study observed approximately a three-point difference in mindfulness scores from baseline to post-test. This difference is consistent with prior research, where changes between two to seven points in mindfulness scores were correlated with improvements in positive mental health [43]. Mothes et al. [44] used a 12-week aerobic exercise intervention and found that regular aerobic exercise increased mindfulness in 149 healthy middle-aged men, while showing a positive impact on mental health. Similarly, a correlational study discovered that individuals who worked out more consistently demonstrated higher mindfulness and acceptance in 266 members of the YMCA [45]. In contrast, another correlational study found no significant association between aerobic exercise and mindfulness [46]. While the relationship is not well-understood, our study posits that aerobic–strength exercise may increase mindfulness over time. In addition to aerobic–strength exercises, yoga is commonly understood to build mindfulness. The practice of yoga and other mindful-based exercises are positively correlated with mindfulness [28]. Furthermore, mindfulness-based interventions have been widely shown to improve overall well-being, including mental and physical health [47]. A study examining 104 regular yoga practitioners aged 18–74 found that higher yoga involvement was related to higher mindfulness and self-compassion [48]. Our study supports previous literature demonstrating the benefits of yoga for mindfulness. In a study similar to ours, comparing the effects of yoga and fitness exercise on stress in 334 undergraduates, yoga was found to have a greater impact on stress due to increased mindfulness, both in the short- and long-term [49]. The present study suggests that both yoga and more strenuous exercise can improve mindfulness. 

In addition to the physical component of our intervention, each group completed supplemental goal-setting and reflective journaling, which may have contributed to the observed increase in mindfulness in both groups. Congruent with this theory, Khramtsova and Glascock [50] found an increase in mindfulness in university students following a journaling and mindfulness intervention. In a repeated measures experiment, Strick and Papiers [51] found that mindfulness facilitated the pursuit of goals in 60 participants (mean age = 21.82 years). This finding of Strick and Papiers sheds light on the findings in our study; perhaps increased mindfulness benefited the act of goal setting, instead of the other way around. Perhaps goal-focused journaling along with mindfulness and/or exercise, per our study’s intervention, is the best way to reap the benefits of each practice. While the current study did not measure success in achieving the goals set, future research should focus on understanding the relationship between goal-setting and reflection as relates to physical activity on the increase of mindfulness scores.

While this study showed promising results for ways to improve mindfulness, neither intervention group had a significant impact on resilience, contrary to our hypothesis. Since previous research indicates that both exercise and mindfulness promote resilience, the reasons for no observed effects of *WeActive* and *WeMindful* on resilience were unexpected. This could be because our study took place during a global pandemic, and perhaps resilience-building is not feasible during a stressful life event. Additionally, our intervention groups only met twice per week and participants were given the flexibility to attend sessions asynchronously to accommodate varying schedules. It is possible that this lack of unity impacted building resilience. Lastly, our intervention was delivered via the university’s Learning Management System, which could have made the intervention feel more like a class than a supportive resource for students. In a meta-analysis examining 268 intervention studies on resilience, Liu et al. [52] found variability in operational definitions of resilience, methodology, and outcomes, as well as significant but small effect sizes, making it difficult to draw definite conclusions from such studies. All in all, our study contradicts the bulk of research showing the benefits of mindful and physical-based exercise on resilience. 

This study had limitations. First, there was no true control group and unequal group sizes, which limits the ability to make generalizable conclusions. However, our analysis examined change from baseline to posttest, as well as examining group differences, which can help illustrate the changes that did occur in this time frame. Further, we were unable to measure participation due to the flexibility of the intervention. This study took place virtually and during lockdown, which likely influenced students’ motivation to participate during the live video sessions. While the intervention was designed to meet the needs of participants at the time, it was difficult to determine adherence and fidelity to the intervention. Additionally, a majority of participants identified as female, potentially skewing results. This could be related to the virtual platform and/or exercise types offered. Finally, the study is limited due to the nature of self-report data which can be subject to bias. In addition, the language used by the instructor may have reinforced terms related to mindfulness and therefore played a role in the increases in mindfulness scores observed in the post-test data. Despite these limitations, this study contributes to the existing literature related to the effects of physical activity intervention on important psychological health outcomes. Future research should also investigate the mechanisms for reaching a mindful state through exercises, comparing aerobic exercise, strength-based exercise, and yoga. Additionally, future research should compare virtual and in-person interventions for physical and mental health during the pandemic. Future interventions should also explore including both a physical and mental component to see effects on college student well-being. While the COVID-19 pandemic-related quarantines have concluded, these findings may be helpful in the case of future pandemics or situations where college students are engaging in their education virtually. Further, web-based physical activity interventions have increased in popularity due to ease of access and scalability, and this research can inform potential impacts.

## 5. Conclusions

Overall, both *WeActive* and *WeMindful* groups, jointly with reflective, goal-setting journaling, all show promise in promoting mindfulness in the college population. However, neither group showed an intervention effect on resilience. The results of our study should guide universities in supporting students’ mental health and promoting access to both exercise and mindfulness resources alike. 

## Figures and Tables

**Figure 1 ijerph-20-05463-f001:**
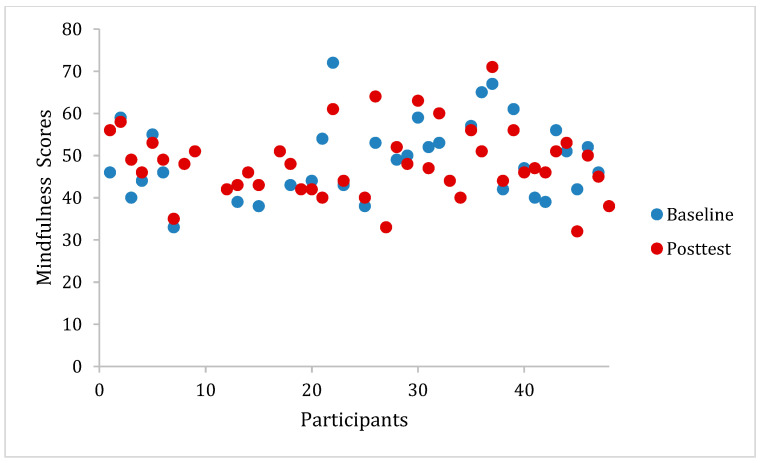
*WeActive* participant responses to the FFMQ-15 at baseline and post-test. Note: FFMQ-15 = Five Facet Mindfulness Questionnaire 15 item scale.

**Figure 2 ijerph-20-05463-f002:**
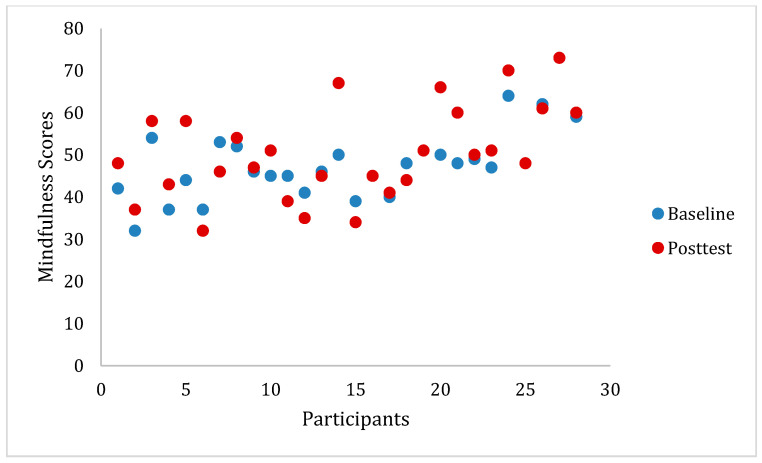
*WeMindful* participant scores on the FFMQ-15 at baseline and post-test. Note: FFMQ-15 = Five Facet Mindfulness Questionnaire 15 item scale.

**Table 1 ijerph-20-05463-t001:** Demographic data of the total participant group.

Variable		*n*	%
Gender	Cisgender female	61	85%
	Cisgender male	7	10%
	Transgender and Gender Non-conforming (TGNC)	4	6%
Race	Asian	14	19%
	Black or African American	4	6%
	White	47	65%
	Multiracial	7	9%
Education status	1st year	7	10%
	2nd year	9	13%
	3rd year	15	21%
	4th year	14	19%
	Master’s	13	18%
	Professional	2	3%
	Doctoral	12	17%

**Table 2 ijerph-20-05463-t002:** Baseline and post-test outcome variable scores by group and for the total sample.

Variable	*WeActive* (*n* = 44)		*WeMindful* (*n* = 28)		Total (*n* = 72)	
	Mean	SD	Mean	SD	Mean	SD
Baseline Resilience	26.48	6.50	24.64	5.50	25.76	6.16
Post-test Resilience	26.80	7.87	26.75	5.54	26.78	7.01
Baseline Mindfulness	33.20	4.74	33.79	5.06	33.43	4.84
Post-test Mindfulness	36.05	5.16	36.61	6.37	36.26	5.62
Baseline Describing	9.07	1.27	9.29	1.36	9.15	1.30
Post-test Describing	9.23	3.03	10.25	3.22	9.63	3.12
Baseline Acting with Awareness	7.95	2.33	8.79	2.36	8.28	2.36
Post-test Acting with Awareness	10.23	2.56	9.46	2.60	9.93	2.59
Baseline Non-Judging	7.64	3.57	7.14	2.90	7.44	3.31
Post-Test Non-Judging	8.27	2.20	7.86	1.20	8.11	2.12
Baseline Non-Reactivity	8.55	2.54	8.57	1.86	8.56	2.28
Post-Test Non-Reactivity	8.32	2.38	9.04	2.58	8.60	2.47

**Table 3 ijerph-20-05463-t003:** Independent t-test for pre-test mean scores between the two groups.

Condition	Group	% or Mean (SD)	t	df	*p*
3x/month exercise	*WeActive*	48%	0.77	59.91	0.443
	*WeMindful*	38%			
3x/month yoga	*WeActive*	17%	−0.81	51.94	4.24
	*WeMindful*	24%			
3x/month therapist	*WeActive*	21%	−2.51	49.15	0.015 *
	*WeMindful*	48%			
Health Rating	*WeActive*	3.14 (1.02)	−0.66	72.29	0.512
	*WeMindful*	3.28 (0.75)			
Baseline Resilience			0.61	68.50	0.545
Baseline Mindfulness			−0.12	71.37	0.905

Note: * = *p* < 0.05. % indicates the percentage of each group that had participated in exercise, yoga, and visited a therapist in the last three months. Health rating is scored from 0 to 5, with 5 representing excellent self-rated health and 1 representing poor self-rated health.

**Table 4 ijerph-20-05463-t004:** Repeated measures ANCOVA results.

Source	Measure	F	*p*	*η^2^*
Time	Resilience	2.56	0.114	0.039
	Mindfulness	5.18	0.026 *	0.070
	Describing	0.02	0.889	0.000
	Acting with Awareness	7.32	0.009 **	0.096
	Non-Judging	5.47	0.022 *	0.073
	Non-Reactivity	0.07	0.796	0.001
Condition	Resilience	0.00	0.97	0.000
	Mindfulness	0.33	0.568	0.005
	Describing	0.44	0.510	0.006
	Acting with Awareness	0.62	0.436	0.009
	Non-Judging	0.83	0.366	0.012
	Non-Reactivity	1.43	0.235	0.020
Time * Condition	Resilience	1.81	0.183	0.026
	Mindfulness	0.00	0.948	0.000
	Describing	0.23	0.635	0.003
	Acting with Awareness	1.04	0.312	0.015
	Non-Judging	0.03	0.870	0.00
	Non-Reactivity	1.42	0.238	0.020

Note: * = *p* < 0.05; ** = *p* ≤ 0.01.

## Data Availability

The datasets generated during and/or analyzed during the current study are available from the corresponding author upon reasonable request.

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
