# Peer review of "Web-Based Physical Activity Interventions to Promote Resilience and Mindfulness Amid the COVID-19 Pandemic: A Pilot Study"

_ijerph, 2023, doi:10.3390/ijerph20085463_

Round 1

Reviewer 1 Report

This study examined whether an online physical activity intervention can improve resilience and mindfulness in college students during a pandemic. Although the initial hypothesis was not supported, unique results were obtained. It will be of interest to readers of this journal. I would like to make a few comments below.

1.       L252. It says three items, but shouldn't it be four items?

2.       What does the horizontal axis in Fig. 1 and 2 represent? Participants? Is this figure necessary in the first place?

3.       Is it appropriate to use a t-test to test the difference in proportions for the 3 items in Table 3 ”3x/month~”? Is the chi-square test better? Also, I am not sure how the Health Rating is coded. Please add a note.

4.       Table4 Acting → Awareness?

5.       Is Fig.3 necessary? Why is resilience the only one shown even though there are no significant results?

6.       I think the lack of a non-interventional control group is a problem. Since this is a pilot study, I would like you to set it appropriately for this test.

Author Response

Thank you so much for reviewing our manuscript. We are so grateful for the time and effort put into the comments. In order for you to know where and how we addressed the comments, I used tracked changes and provided a line location in the response. Further, I providef a point-to-point responses to the specific comments.

  1. It says three items, but shouldn't it be four items?

According to the FFMQ-15, there are three items per facet (3 items x 5 facets = 15 total items).

  1. What does the horizontal axis in Fig. 1 and 2 represent? Participants? Is this figure necessary in the first place?

Yes, it represents participants. We have added an x-axis label, please see page 9.

  1. Is it appropriate to use a t-test to test the difference in proportions for the 3 items in Table 3,”3x/month~”? Is the chi-square test better? Also, I am not sure how the Health Rating is coded. Please add a note.

We believe a t-test is as appropriate as a chi-square, given that we are examining group differences. We have added information on the health rating, please see page 10, line 375-376.  

  1. Table4 Acting → Awareness?

Thanks for this comment. We have fixed the label in the table on pages 8-9.

  1. Is Fig.3 necessary? Why is resilience the only one shown even though there are no significant results?

We have removed Figure 3 in accordance with your comment.

  1. I think the lack of a non-interventional control group is a problem. Since this is a pilot study, I would like you to set it appropriately for this test.

Thank you for your comment. We understand that the lack of a non-intervention control is a limitation in our study. By examining changes from baseline to posttest, as well as examining group differences, we are able to understand the changes that did occur without a control group. This pilot study helped show how a virtual intervention may be effective in supporting college students’ mental health. We note this as a limitation in our study on page 12, lines 457-460.

Reviewer 2 Report

Dear authors,
Thank you for your paper named "Web-Based Physical Activity Interventions to Promote Resilience and Mindfulness amid the COVID-19 Pandemic: A Pilot Study". It is a nicely conducted pilot study to address the issue regarding web-based physical activity interventions as the potential solution to promote mindfulness during the pandemic.

Author Response

Thank you so much for reviewing our manuscript. We are so grateful for your positive review of the manuscript. We appreciate your time and efforts in this revision.

Reviewer 3 Report

Dear authors,

Manuscript shown the evidence of using WeActive and WeMindful in college students during a 8 weeks period. I have some considerations for you.

Abstract: a short introduction to explain the topic of the manuscript should be added before the purpose of the study.

Introduction: complete and with a lot of references that support your purpose.

Methodology:

Line 120-124: this information should be placed at results section.

Line 148: WeActive. Is there any communication between the user and the teacher? Maybe, in case of doubt or the inability to perform any exercise?

Line 167: How was the feedback of the participants? Is there any specific questionnaire, free answer questions? Please explain that point.

Line 171: are there any theoretical lessons? In case of negative answer, did you implement a control about the frequency the exercise/lessons were done? Because you recommend/establish 2 of 30 min, but you also mention that the users could visualize lesson at any time they want. It looks a bit contradictory.

Line 195: are this lesson mandatory or voluntary?

Line 227: outcome measures. Personally I prefer to place this section before procedure, to improve the reader comprehension.

Line 265-269: this information corresponds to data results.

Line 280-287: sample size calculation should be places in section 2.1.

Results.

Table 1 should be here.

I would have liked to read the results of the feedback obtained from the coaching sessions as well as the personal impressions of the users during these weeks.

Discussion:

Like I have mention before, some results are not explaining in discussion section. Is there another app which combining both issues (physical and mental)? It could be proposed as a future investigation line?

Author Response

Thank you so much for reviewing our manuscript. We are so grateful for the time and effort put into the comments. In order for you to know where and how we addressed the comments, I used tracked changes and provided a line location in the response. Further, I will provide a point-to-point responses to the specific comments.

Abstract

  1. A short introduction to explain the topic of the manuscript should be added before the purpose of the study.

Thanks for your comment. We added a few sentences to explain the topic in the abstract, please see page 1, lines 10-12.

Introduction

  1. Complete and with a lot of references that support your purpose.

Methodology:

  1. Line 120-124: this information should be placed at results section.

Thanks for your comment. We have moved this to the results section, please see page 8, lines 339-342.

  1. Line 148: WeActive. Is there any communication between the user and the teacher? Maybe, in case of doubt or the inability to perform any exercise?

Thanks for this question. The users were able to ask the instructor via the chat function in Zoom or via email if they had questions or needed modifications in real-time. We have added this information to the manuscript, please see page 5, lines 211-212.

  1. Line 167: How was the feedback of the participants? Is there any specific questionnaire, free answer questions? Please explain that point.

Thanks for your comment. We have added information on our feedback of participants. Page 5, lines 267-270, explain our feedback process.

  1. Line 171: are there any theoretical lessons? In case of negative answer, did you implement a control about the frequency the exercise/lessons were done? Because you recommend/establish 2 of 30 min, but you also mention that the users could visualize lesson at any time they want. It looks a bit contradictory.

We did not provide any theoretical lessons. One of the challenges and limitations of providing a flexible and accessible intervention is the lack of fidelity data, so we weren’t able to track exact physical activity minutes. We realize this is a limitation of our study, and have this noted in the discussion, page 12, lines 463-465.

Line 195: are this lesson mandatory or voluntary?

We encouraged participants to attend these lessons as part of the intervention, but we wanted to leave flexibility for participants given the state of the pandemic. As noted above, it was difficult to ascertain fidelity data.

  1. Line 227: outcome measures. Personally, I prefer to place this section before procedure, to improve the reader comprehension.

Thanks for this comment. We have moved the outcome measures to before the intervention conditions. Please see page 4, lines 155-191.

  1. Line 265-269: this information corresponds to data results.

We have moved this information to the results along with the demographic information. Please see page 8, lines 339-349

  1. Line 280-287: sample size calculation should be places in section 2.1.

Thank you, this has been moved to 2.1, please see page 3 lines 136-143.

Results

  1. Table 1 should be here.

We have moved Table 1 to the results, please see page 8.

  1. I would have liked to read the results of the feedback obtained from the coaching sessions as well as the personal impressions of the users during these weeks.

We did not record individual responses to the feedback session. We wanted the participants to respond honestly, so we did not record the session. However, overall themes were noted by the researchers, and we made appropriate changes to the intervention, as noted on page 6, lines 267-270.

Discussion

12. Like I have mention before, some results are not explaining in discussion section. Is there another app which combining both issues (physical and mental)? It could be proposed as a future investigation line?

Thanks for this comment. We have added information to the discussion to address this. Please see page 12 lines 475-476.

Reviewer 4 Report

Minor issues

- Please remove the info regarding the participants from data analysis and transfer them into the participants’ section.

- My main concern is that why do you not include a control group into the study? Please clarify.

- Due to the fact that covid-19-related quarantine is almost over, the suggestions for the results of this stud should be related to possible future pandemics. Please reconsider the suggestions.

Author Response

Thank you so much for reviewing our manuscript. We are so grateful for the time and effort put into the comments. In order for you to know where and how we addressed the comments, I used tracked changes and provided a line location in the response. Further, I will provide a point-to-point responses to the specific comments.

Reviewer #4:

  1. Please remove the info regarding the participants from data analysis and transfer them into the participants’ section.

Thanks for your comment. We have moved the participant demographic info and the data analysis section to the results based on your comments and comments from other reviewers. Please see page 8, lines 339-350.

  1. My main concern is that why do you not include a control group into the study? Please clarify.

We understand that the lack of a non-intervention control is a limitation in our study. By examining changes from baseline to posttest, as well as examining group differences, we are able to understand the changes that did occur without a control group. This pilot study helped show how a virtual intervention may be effective in supporting college students mental health. We have added information in the discussion, please see page 12, lines 457-460.

  1. Due to the fact that covid-19-related quarantine is almost over, the suggestions for the results of this stud should be related to possible future pandemics. Please reconsider the suggestions.

Thank you for this comment. We have added a few sentences at the end of the discussion addressing this. Please see page 12 lines 475 to 481.

Round 2

Reviewer 3 Report

Dear authors, 

Thank you, manuscripts looks much better.